# Communications Is Time for Care: An Italian Monocentric Survey on Human Papillomavirus (HPV) Risk Information as Part of Cervical Cancer Screening

**DOI:** 10.3390/jpm12091387

**Published:** 2022-08-26

**Authors:** Miriam Dellino, Eliano Cascardi, Valentina Tomasone, Rosanna Zaccaro, Katia Maggipinto, Maria Elena Giacomino, Miriana De Nicolò, Simona De Summa, Gerardo Cazzato, Salvatore Scacco, Antonio Malvasi, Vincenzo Pinto, Ettore Cicinelli, Carmine Carriero, Giovanni Di Vagno, Gennaro Cormio, Chiara Antonia Genco

**Affiliations:** 1Department of Biomedical Sciences and Human Oncology, University of Bari, 70121 Bari, Italy; 2Clinic of Obstetrics and Gynecology, “San Paolo” Hospital, 70121 Bari, Italy; 3Department of Medical Sciences, University of Turin, 10124 Turin, Italy; 4Pathology Unit, FPO-IRCCS Candiolo Cancer Institute, Str. Provinciale 142 km 3.95, 10060 Candiolo, Italy; 5Bioinformatician-Molecular Diagnostics and Pharmacogenetics Unit, IRCCS-Istituto Tumori “Giovanni Paolo II”, 70124 Bari, Italy; 6Department of Emergency and Organ Transplantation, University of Bari “Aldo Moro”, 70121 Bari, Italy; 7Department of Basic Medical Sciences and Neurosciences, University of Bari “Aldo Moro”, 70121 Bari, Italy; 8Gynecologic Oncology Unit, IRCCS Istituto Tumori Giovanni Paolo II, Department of Interdisciplinary Medicine (DIM), University of Bari “Aldo Moro”, 70121 Bari, Italy; 9Departmental of Cervical-Carcinoma Screening, ASL Bari, 70121 Bari, Italy

**Keywords:** precision medicine, health promotion, cervical cancer, cervical lesion, screening, HPV, CIN

## Abstract

Human papilloma virus (HPV) infection could be considered a social disease, both for its high incidence, especially in younger subjects, and for the risk of neoplastic evolution linked to viral infection. Therefore, the National Health System, in collaboration with the state, must help women to understand the oncological risk of HPV and suitable methods of prevention. We conducted an Italian monocentric survey on HPV risk information as part of cervical cancer screening. An anonymous questionnaire was administered to 200 women with high-risk positive HPV and low-grade cervical lesions during second-level cervical cancer screening at the Gynecology and Obstetrics Unit of the “San Paolo” Hospital. From this survey, the need to improve communication for patients has emerged, as currently it is not exhaustive. In response to this need, organizational changes have been implemented to centralize the moment of counseling in the second levels of screening and to improve the training of health workers in level I as well as family doctors. In addition, psychological support was also proposed to patients who requested it, as was the dissemination of material such as that produced by GISCI (Italian Cervico-Carcinoma Screening Group) and updated in May 2018, which provides 100 answers to questions on HPV in order to achieve effective and comprehensive communication. This investigation requires further development, and the expansion of this investigation to the multicenter level is already underway. Therefore, this survey will represent a cornerstone for further discussion on the topic considering the necessity of appropriate communication in the oncological context.

## 1. Introduction

Human papillomavirus (HPV) infection is widely described as the most common virus in the uterine cervix cells, and high oncogenic risk genotypes 16 and 18 are the main players involved in tumor progression with a percentage of over 70% in the literature [1]. Furthermore, these genotypes, regardless of gender, represent a mechanism by which tumors or cancerous processes can develop in the genital area, as found in several cases of benign and malignant forms of the anus, penis and vulva [2], which very often require an early differential diagnosis [3,4,5]. Currently, cervical carcinoma is considered one of the most common cancers in women, with a mortality rate of 7.5% in oncologic patients [6,7]. Nonetheless, most HPV-related lesions are reversible if promptly detected in precancerous stages during first-level screening with HPV-DNA test [8] as suggested by the World Health Organization [9] and in agreement with others guidelines [10,11,12,13]. In addition to proper prevention through screening, the World Health Organization recommends and promotes the use of vaccines as a means of preventing the onset of cervical cancer, as is already in use in many states [14]. In this regard, the first anti-HPV vaccination campaign was registered in Italy in 2007 when the vaccine was also offered to 11-year-old girls. After five years, there was vaccination coverage of about 69%, while the data from four years ago for women born in 2006 show coverage of 40% for the complete vaccination course [15]. These data not only show that “giant steps” have been made, but that further efforts are needed to achieve full awareness of this disease among women. It is therefore very important to insist and inform people that cervical cancer is a preventable and treatable disease and to communicate that certain measures are available for its prevention and early diagnosis. Indeed, it is the opinion of many that even today, women may encounter considerable difficulties and obstacles in obtaining information or access routes relating to HPV and its prevention, and the possibility that specific or environmental factors [8] may influence these information gaps needs to be addressed. These assessments should therefore be contemplated in the HPV information stages considering that: (i) high-risk strains are sexually transmitted [16], (ii) HPV awareness levels among women are low [15,17,18], and (iii) patients often have unanswered questions [19,20] and want more information on areas such as viral types and implications for sexual relationships [21,22]. Furthermore, the success of vaccination programs is dependent on a high level of dissemination, which is notoriously influenced by the media on the principles of vaccination, the views of the family circle and individual perceptions Therefore, considering that this issue certainly has a global implication in consideration of the fight against vaccination abstention, this issue must be addressed with extreme clarity [23] using written health communication media [24] as clear and effective information tools for helping patients understand the key role of HPV in cervical cancer. Health literacy is “the degree to which individuals have the capacity to obtain, process, and understand basic health information” [25]. Lower health literacy levels are associated with lower vaccination adoption rates and cancer screening adoption [26]. For this reason, health information should be legible regardless of literacy level, presenting widely understandable written texts [27] that do not adversely affect the understanding or behavior of individuals regarding the prevention, early detection and early treatment of cervical cancer [24]. In this regard, health professionals should develop easy-to-read health information on HPV vaccination, screening and cervical cancer, as well as watch over non-accredited sources which, however, can be an easy way of communicating for patients [25]. Nowadays, social media are considered an ideal tool for obtaining information on vaccines as compared to newspapers or television; they can be more quickly accessible and in some cases even provide customizable news that can also be discussed and evaluated among different users in virtual communities with the increased risk of receiving incomplete or incorrect information [26]. The aim of our study was to explore women’s information sources and needs, collect their suggestions to improve communication following HPV+/Cytology and consequently actuate specific changes to the current organization of screening.

## 2. Materials and Methods

An anonymous questionnaire (Appendix A) was administered to 200 women with high-risk positive HPV and low-grade cervical lesions during the second-level cervical cancer-screening center of the Gynecology and Obstetrics Unit of the “San Paolo” Hospital. The questionnaire included 19 dual-answer questions (yes or no), and a final open-ended question. This questionnaire was formulated to investigate knowledge and sources of information about HPV and its role in cervical cancer. All data were reported in an Excel file. Each patient was informed about the aim of the study and signed informed consent before allowing data collection for research purposes. All stages of research were conformed to the Helsinki Declaration (1964 and later versions) and to RECORD (Reporting of Studies Conducted using Observational Routinely-collected Data) statement [28]. These data were collected during routine clinical activity and made anonymous. The favorable opinion of the Institutional Review Committee is therefore not essential. To compare the frequencies of response to every question, the chi-square test was performed, stratifying patients according to their response to question 4. Both univariate and multivariate logistic regression identified independent variables associated with responses to question 4.

## 3. Results

The main patients’ clinical characteristics are reported in Table 1.

Patients were stratified according to their response to question 4 and through the chi-square test; the remaining answers were compared. The result is statistically significant (*p*-value < 2 × 10^−16^), with answering “no” to questions 1, 2, 3 and 16 being strongly associated with the group of patients that found themselves to be not appropriately informed about HPV (Figure 1).

It could also be observed that knowledge about HPV (84.5, 95%CI: 15.8–1570) and colposcopy (21.4, 95%CI: 7.49–71.7) and the presence of a clinician able to provide extensive information (7.93, 95%CI: 3.01–21.4) increased the chance that a patient was appropriately informed (Table 2).

Thus, a multivariate logistic model was fitted considering the above-mentioned variables. The independent variables were the knowledge of colposcopy and the amount of information provided by the clinician (Figure 2).

## 4. Discussion

HPV screening is available to the entire population as a first-level test within territorial health structures [29,30].

The HPV test has a higher detection rate compared to cytology, which is now performed as a second step in patients identified as being high-risk (about 6%–12% depending on the prevalence of infection) [27,31,32,33,34]. As a consequence, the number of cytology readings decreases, but the probability of observing abnormal ones among those examined increases by more than 10 times (in Italian studies, this was shown to be between 20% and 55%) [34,35,36,37,38]. In the case of positivity of the first-level test, these patients collect the result of the HPV test at the clinic where they receive information relating to the outcome of the HPV test and are sent to the level II or III centers for further investigation [14,39,40,41,42]. The survey carried out at our center showed that only 12% of patients belonging to our level II and III centers felt uninformed (Question 16), as 36% had acquired information from the media (Question 8). Of these, in fact, 38.5% reported that they had not felt reassured by the information collected independently from the internet (Question 13). Therefore, 48% reported the need for comprehensive and standardized communication by qualified personnel (Question 17). In addition, 63% of patients reported (Question 15) that the use of the terms “High Risk” and “Low Risk” in the current reporting system increased their feelings of anguish and anxiety. The data relating to how (Question 18) 13% of patients have resorted to psychological support and 34.5% believe it is necessary to implement this as part of the screening process as a support to women with HPV lesions is important. In this regard, the recent operating model in force since 01/09/22 and presented following Resolution of the Regional Council No. 748 of 23/05/2022 n.1332/2020 regarding “Organization of cancer screening in the Puglia region—Operational indications—Cervical cancer screening program—Operational Protocol and transition to the HPV DNA Test” has made important changes [43,44,45], in particular, delegating to the level II center the explanation of the positive report as well as the execution of the specialized study [26,27,28]. This could allow women to have a reference or unique team, composed of accredited and dedicated personnel in the field of cervical cancer screening [46,47].

In addition, there could be further integration of communication through the posting of iconographic material, to be posted in the dedicated screening centers both to better explain the evolution of the HPV-related dysplastic pathology (Figure 3) and to show the tools available to eradicate the infection (Figure 4).

In addition, to date there are national and official information systems, provided by the GISCI (Italian Cervico-Carcinoma Screening Group) and updated in May 2018, which illustrate the collection of 100 answers to questions on HPV, whose content could be disseminated both on paper and electronically [48,49]. In addition, the new resolution described the need to carry out training courses for personnel present at the territorial level of level I, as well as for family doctors, who often represent the first point of reference for the patient. In fact, in the field of prevention, the sooner information is communicated, the more effective it is [50]. Therefore, training programs should be established within schools for both pupils and parents to implement adherence to preventive programs and vaccination campaigns. In Puglia in 2019, adherence to cervical cancer screening was 31.9% and therefore lower than the national average (88% in the north and 69% in the south). On the other hand, the Australian model demonstrates how implementing communication improves patient compliance [51]. In fact, in Australia a campaign to promote HPV vaccination has been heavily funded since 2007 in girls and subsequently in adolescents since 2013, with adherence rates to primary prevention among the highest observed throughout the world [51]. Substantial declines in high-grade cervical disease and genital warts have been observed among women who have had the HPV vaccine [51]. In addition, a reduction in the incidence of genital warts and HPV-related genital dysplastic lesions has been reported among heterosexual men [51]. However, significant vaccination adhesions have also been recorded in other states, demonstrating how this problem has worldwide relevance [52,53,54,55,56]. In fact, the 9vHPV vaccine is expected to prevent up to 90% of cervical cancers and 96% of cancers. Of about 1544 cancers associated with HPV in 2012, 1242 were preventable by the 4vHPV vaccine and another 187 anogenital cancers by the 9vHPV vaccine [51]. Therefore, the transition to 9vHPV will further reduce the onset of dysplasia and neoplasia associated with HPV, with a continuous high coverage among both males and females and with the possibility of completely eradicating the virus [51]. For this reason, in addition to being standardized on the notions to be provided to the patient so that they are clear and consistent, doctors and paramedical staff should make use not only of tools such as iconographic material within the clinics to make counseling effective, but also paper or computer distribution tools, in order to allow effective and exhaustive communication that consciously reassures the patient [48,57], as “communication is time for care”, as well as a fundamental and necessary requirement in the field of cancer prevention.

## 5. Conclusions and Future Direction

The care of communication with the patient in medicine is fundamental. Therefore, the use of tools to facilitate understanding of the oncogenic risk related to HPV infection can change the evolution of the disease. Consequently, its use must be sensitized, and doctors and paramedical staff must be adequately trained. Consequently, this study represents a first step to bring back “the voice of patients”, which expresses the need for greater care in communication, and new projects may arise. In particular, this document could be a valid support tool in schools and educational centres for sensitizing women, mothers and daughters to this problem by informing them of the opportunities available to them or encouraging them to learn more. Indeed, vaccine prevention in women who have not yet had sexual intercourse may hold the key to eradicating HPV-related cervical cancer. With the same objective, it could be an integral part of brochures or information material to be distributed during awareness campaigns or days dedicated to HPV screening. Finally, the possibility should also be considered that this questionnaire could be a useful tool as psychological support within medical clinics or could even become a dissemination tool through the creation of dedicated and controlled online pages or institutional communications.

## Figures and Tables

**Figure 1 jpm-12-01387-f001:**
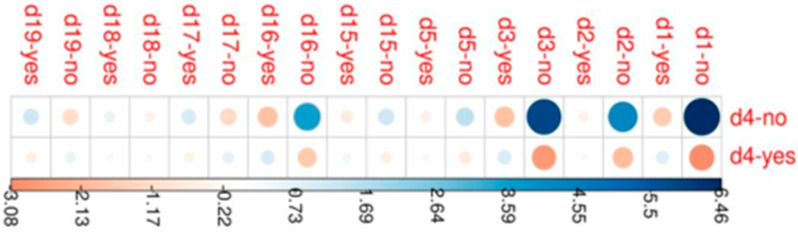
Correlation plot depicting Pearson residual related to response to question 4. Univariate logistic regression was performed to identify which of the parameters, evaluated by the questionnaire, are able to influence appropriate communication on Human Papillomavirus risks.

**Figure 2 jpm-12-01387-f002:**
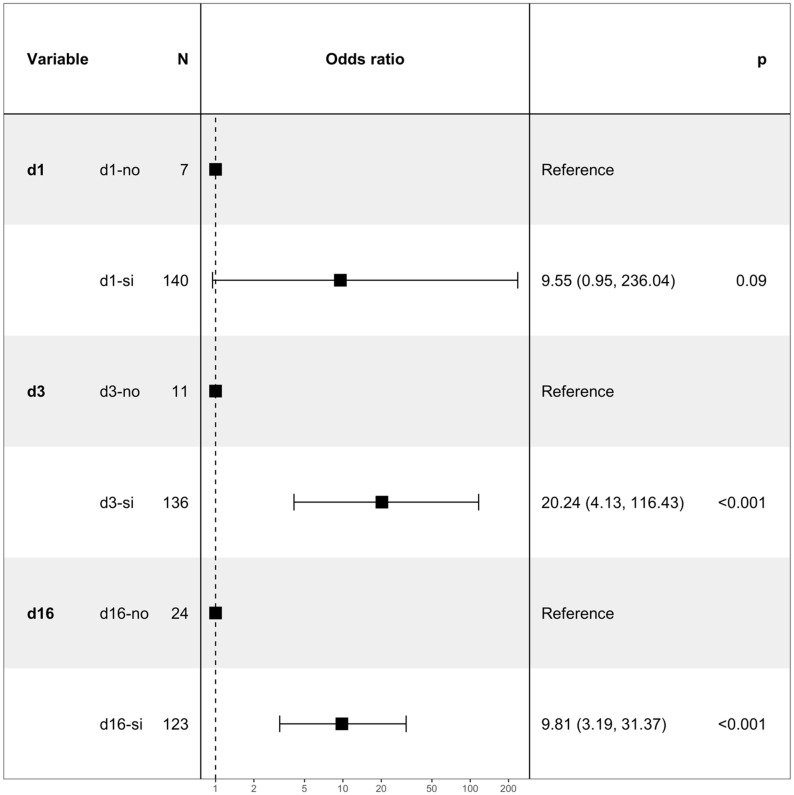
Forest plot related to multivariate logistic regression.

**Figure 3 jpm-12-01387-f003:**
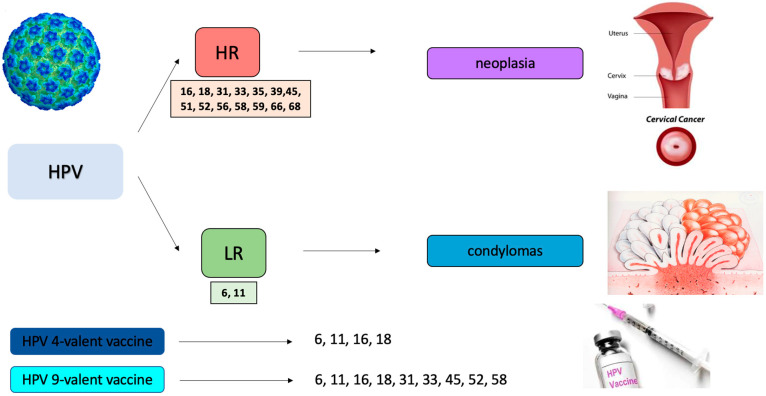
Example image of the etiopathogenesis of HPV to optimize counseling with the patient (HR: high risk; LR: low risk).

**Figure 4 jpm-12-01387-f004:**
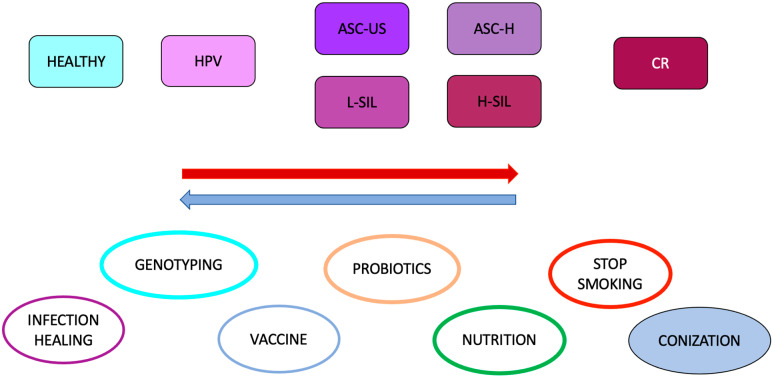
Example image of the tools currently available to counteract the pathogenesis of HPV-related infection (ASC-US: atypical squamous cells of undetermined significance, ASC-H: atypical squamous cells cannot exclude H-SIL, L-SIL: low-grade squamous intraepithelial lesions, H-SIL: high-grade squamous intraepithelial lesions, CR: cervical carcinoma).

**Table 1 jpm-12-01387-t001:** Features and risk factors of HPV-positive patients enrolled in the study.

Characteristic	Data	Number of Patients (%)
Age (median years)	45 (30–64)	/
Ethnic origin	Caucasian	100
History of cervical cancer	14/200	7
Educational level	Advanced: 38/200	19
	Intermediate: 74/200	37
	Basic: 76/200	38
	Less than basic: 12/200	6
Income level	Low income: 80/200	40
	Lower middle income: 78/200	39
	Upper-middle income: 33/200	16.5
	High income: 9/200	4.5
Marital status	Married: 54/200	27
	Single: 79/200	39.5
	Divorced: 67/200	33.5
Number of sex partners	7.8 (±6.5)	/
Nullipara	80/200	40
Pluripara	120/200	60
Monogamous heterosexual relationship	124/200	62
Same-sex relationships	36/200	18
Smoking	72/200	36
Contraceptive pill use	38/200	19
History of infection with other sexually transmitted infections	63/200	31.5

**Table 2 jpm-12-01387-t002:** Univariate logistic regression results (1OR = Odds Ratio, CI = Confidence Interval).

Characteristic	N	OR1	95% CI1	*p*-Value
D1	194			
no		—	—	
yes		84.5	15.8, 1,570	<0.001
D2	194			
no		—	—	
yes		76,789,489	0.00, NA	>0.9
D3	191			
no		—	—	
yes		21.4	7.49, 71.7	<0.001
D5	193			
no		—	—	
yes		2.02	0.89, 4.41	0.084
D15	151			
no		—	—	
yes		2.20	0.77, 5.85	0.12
D16	149			
no		—	—	
yes		7.93	3.01, 21.4	<0.001
D17	144			
no		—	—	
yes		0.00		>0.9
D18	152			
no		—	—	
yes		0.67	0.25, 2.00	0.4
D19	147			
no		—	—	
yes		0.45	0.18, 1.05	0.069

## Data Availability

All results are reported within the text.

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
