# Peer review of "Communications Is Time for Care: An Italian Monocentric Survey on Human Papillomavirus (HPV) Risk Information as Part of Cervical Cancer Screening"

_jpm, 2022, doi:10.3390/jpm12091387_

Round 1

Reviewer 1 Report

The authors in the manuscript entitled " Communications is time fore care: an Italian monocentric survey on Human Papillomavirus (HPV) risk information as part of cervical cancer screening" was discussed about HPV and Cervical cancer which accounted for 7.5% of all female cancer fatalities . Most of the cases are treatable if detected early by screening. WHO recommendation comes along with several regions as they have authorized HPV vaccinations . Most cervical cancers are prevented; therefor, this work could provide a guide for patients for more cancer prevention.

I suggest for the authors to clarify the so many sentences grammatically. There are several clear mistakes. 

I suggest to add a separate  conclusion and future direction parts to help readers and researchers use this manuscript as example for HPV cancer prevention. 

I suggest to include more references in discussion related to countries who implemented HPV vaccinations as part of their women health system.

Overall, the manuscript is well written. 

Author Response

The answer is attached.

Reviewer 2 Report

The study was designed to investigate the importance of communication to help patients understand the cancerous risk of HPV infection in Italy.  The study is timely and critical because cervical cancers associated with HPV infection is preventable but the rate of completion of HPV vaccination in Italy remains low.

However, there are several major points that were not considered by the authors.

1.       The authors need to provide the age as well as other demographic information of the 200 subjects. The addition of a table providing the above-mentioned information is a must.

2.       Age, education level, income level, occupation, family history of cervical cancers, marital status, number of sex partners, history of infection with other STIs, need to be included in the data analysis. These are all risk factors associated with HPV infection and therefore potentially affect how the patients answer the 20 questions designed by the authors.     

Author Response

The answer is attached.

Round 2

Reviewer 2 Report

Table 1, the authors need to carefully check the "number of patients%): under educational levels the calculation was not correct (check intermediate and less than basic).

Author Response

Thank you for identifying the error which was promptly corrected.